# Reductive cyanation of organic chlorides using CO$_2$ and NH$_3$ via Triphos–Ni(I) species

Yanan Dong[1,2], Peiju Yang[1], Shizhen Zhao[3] & Yuehui Li [1]✉

Cyano-containing compounds constitute important pharmaceuticals, agrochemicals and organic materials. Traditional cyanation methods often rely on the use of toxic metal cyanides which have serious disposal, storage and transportation issues. Therefore, there is an increasing need to develop general and efficient catalytic methods for cyanide-free production of nitriles. Here we report the reductive cyanation of organic chlorides using CO$_2$/NH$_3$ as the electrophilic CN source. The use of tridentate phosphine ligand Triphos allows for the nickel-catalyzed cyanation of a broad array of aryl and aliphatic chlorides to produce the desired nitrile products in good yields, and with excellent functional group tolerance. Cheap and bench-stable urea was also shown as suitable CN source, suggesting promising application potential. Mechanistic studies imply that Triphos-Ni(I) species are responsible for the reductive C-C coupling approach involving isocyanate intermediates. This method expands the application potential of reductive cyanation in the synthesis of functionalized nitrile compounds under cyanide-free conditions, which is valuable for safe synthesis of (isotope-labeled) drugs.

[1] State Key Laboratory for Oxo Synthesis and Selective Oxidation, Suzhou Research Institute of LICP, Center for Excellence in Molecular Synthesis, Lanzhou Institute of Chemical Physics (LICP), Chinese Academy of Sciences, 730000 Lanzhou, P. R. China. [2] University of Chinese Academy of Sciences, 100049 Beijing, P. R. China. [3] Key Laboratory of Receptors-Mediated Gene Regulation and Drug Discovery, School of Medicine, Henan University, 475001 Kaifeng, P. R. China. ✉email: yhli@licp.cas.cn

Nitriles are key intermediates in production of pharmaceuticals, agrochemicals, organic materials, and functional materials[1,2]. Moreover, nitrile group serves as a versatile motif for the synthesis of aldehydes, ketones, carboxylic acids, alcohols, amides, amines, and heterocycles[3]. Significant progress has been made in transition-metal-catalyzed cyanation in the past century, among which cyanation of aryl (pseudo)halides is the most commonly used approach for selective synthesis of aryl nitriles[4–7].

Careful selection of cyanating reagents is usually required. In this regard, the direct use of metal/silyl cyanides or the in situ generation of cyanide ion is often involved (Fig. 1a, b: $K_4[Fe(CN)_6]$[8–14], $K_3[Fe(CN)_6]$[15], ethyl cyanoacetate[16,17], acetone cyanohydrin[18–20], butyronitrile[21], 4-cyanopyridine N-oxide[22], formamide[23], $DMF/NH_4^+$[24–26], etc). These reactions proceed through Rosenmund-von Braun reaction mechanism and suffer from limited catalytic efficiency and/or substrate scope due to strong coordination ability and poisoning of cyanide ion.

Meanwhile, elegant development of cyanide-free methods involving diverse mechanisms to circumvent this challenge has been achieved including β-H elimination-facilitated cyanation[27,28], electrophilic cyanation with N-cyano-N-phenyl-p-methylbenzenesulfonamide[29], oxidative cyanation with tert-butyl isocyanide[30,31] or hexamethylenetetramine[32] etc. Besides, the recently emerging cyanation under reductive conditions as well undergoes different mechanisms, and in principle offers a complementary synthesis protocol for nitriles. Pioneering work by Cheng and co-workers used Pd/Ni-phosphine complexes for the catalytic cyanation of aryl halides with acetonitrile in the presence of zinc powder at 160 °C[33]. Also using Zn as reductant, Rousseaux and co-workers applied electrophilic cyanating reagent 2-methyl-2-phenyl malononitrile for the cyanation of aryl halides[34]. Very recently, Tsurugi and Mashima et al. developed the use of acetonitrile as cyano source using N-silylated dihydropyrazines as reductant[35]. However, to the best of our knowledge none of these systems are suitable for general cyanation of the least expensive and most widely available organic chlorides having the relatively inert $C(sp^2)$-Cl bond (ca. 96 kcal mol$^{-1}$). Therefore, there is an increasing need to develop general methods for cyanide-free cyanation of organic chlorides.

In order to synthesize nitriles through reductive cyanation pathway, the use of higher oxidation state $CO_2$ as the carbon source is ideal for charge balance purpose. Accordingly, there are challenges to be overcome: (1) suppressing reductive dehalogenation; (2) suppressing the undesired $CO_2$ reduction; 3) conquering possible formation of stable urea, and (4) generation of active electrophile ready for C–C bond formation and C–N triple bond construction (Fig. 2). Accordingly, the choice of metal precursors and ligands is crucial to suppress the above mentioned side-reactions.

The discovery of cyanation reagents with operational simplicity and economic viability is a sought-after goal in organic synthesis. In this respect, CN source from sustainable C1 and N1 feedstocks is a natural line of enquiries. The fixation and transformation of carbon dioxide ($CO_2$) has attracted considerable attention[36–40]. Carbon dioxide is the most abundant carbon source, and is thus regarded as the most promising nontoxic C1 feedstock[41–46]. Recently, we developed the Cu-catalyzed reductive cyanation of aryl iodides with $CO_2/NH_3$ as cyano (CN) source[47]. Herein, we report the reductive cyanation of organic chlorides using $CO_2/NH_3$ or urea as either gaseous or solid forms of carbon and nitrogen, respectively (Fig. 1c).

## Results

**Reaction discovery.** Our study began by investigating catalytic cyanation of chlorobenzene with ambient $CO_2$ and $NH_3$ in the presence of PhSiH₃ and KF (Table 1). The model reaction was firstly tested with different metal catalysts (also see Supplementary Table 1). Among a wide range of catalysts examined, only nickel salts provided the desired product (Table 1, entries 1–5). Further reaction optimization indicated that $Ni(acac)_2$ was a superior choice (entry 6). The addition of Zn powder as coreductant promoted the reaction significantly raising the yield to 37% (Table 1, entry 7). Gratifyingly, evaluation of ligands showed remarkable performance of Triphos for this transformation to produce **2a** in 81% yield (Table 1, entries 8–13). Screening of solvents and silane reductants were carried out. Other high boiling point polar solvents like DMI were found less suitable (Table 1, entry 14). Notably, the choice of silanes also appeared to be crucial for the reaction with PMHS giving poor reactivity (Table 1, entry 15).

**Previous works:**

**a** metal cyano source

**b** organic cyano source

**This work:**

**c** CN source for complex nitriles

■ aryl and aliphatic chlorides ■ many functional groups tolerated

■ safe and cyanide-free ■ cheap and abundant $CO_2/NH_3$

**Fig. 1 Catalytic cyanations with different cyano sources. a** The use of metal cyanides, **b** organic cyano compounds, and **c** $CO_2/NH_3$ as the CN source.

**a** reductive dehalogenation

**b** reduction of $CO_2$ and waste of reductant

**c** generation of highly stable urea

**d** $CO_2/NH_3$ as the electrophile: formation of C-C and C≡N

**Fig. 2 Challenges for reductive cyanation strategy. a** Reductive dehalogenation. **b** reduction of $CO_2$. **c** The generation of stable urea. **d** The challenging generation of active electrophiles.

**Table 1 Optimization of the reaction conditions[a].**

| Entry | Cat. | Ligand | Yield (%) |
|---|---|---|---|
| 1[b] | Pd(OAc)$_2$ | dppp | n.d. |
| 2 | Cu(OAc)$_2$ | dppp | n.d. |
| 3 | Co(acac)$_2$ | dppp | n.d. |
| 4 | Ni(COD)$_2$ | dppp | 2 |
| 5 | NiBr$_2$ | dppp | 17 |
| 6 | Ni(acac)$_2$ | dppp | 20 |
| 7[c] | Ni(acac)$_2$ | dppp | 37 |
| 8[c] | Ni(acac)$_2$ | dppe | 4 |
| 9[c] | Ni(acac)$_2$ | dppb | 5 |
| 10[c] | Ni(acac)$_2$ | dppf | 10 |
| 11[c] | Ni(acac)$_2$ | PPh$_3$ | n.d. |
| 12[c] | Ni(acac)$_2$ | Tripod | 64 |
| 13[c] | Ni(acac)$_2$ | Triphos | 81 |
| 14[c,d] | Ni(acac)$_2$ | Triphos | 16 |
| 15[c,e] | Ni(acac)$_2$ | Triphos | 5 |

*NMP* 1-methyl-2-pyrrolidinone, *DMI* 1,3-dimethyl-2-imidazolidinone, *acac* acetylacetonate, *COD* 1,5-cyclooctadiene, *dppp* 1,3-bis(diphenylphosphino)propane, *dppe* 1,2-bis(diphenylphosphino)ethane, *dppb* 1,4-bis(diphenylphosphino)butane, *dppf* 1,1′-bis(diphenylphosphino)ferrocene, *Tripod* 1,1,1-Tris(diphenylphosphinomethyl)ethane, *Triphos* bis(2-diphenylphosphinoethyl)phenylphosphine, *PMHS* poly (methylhydrosiloxane), *n.d.* not detected.
[a]Reaction conditions: **1a** (0.125 mmol), CO$_2$/NH$_3$ (1/1 atm), NMP (0.5 mL), 20 h; GC yield.
[b]5 mol% Pd(OAc)$_2$.
[c]With 1.0 equiv. of Zn powder.
[d]DMI as solvent.
[e]PMHS was used instead of PhSiH$_3$.

**Cyanation using CO$_2$/NH$_3$ and the synthetic application.** Encouraged by the results above, we then investigated the utility and scope of this method. An array of aryl chlorides bearing various substituents were well-tolerated to afford the desired nitrile products in moderate to high yields (Fig. 3). [13]C-labeled benzonitrile [13]C-**2a** was conveniently obtained using [13]CO$_2$ (83% yield), which confirmed that the carbon source of cyano moiety was derived from CO$_2$ (see Supplementary Figs 2, 3). We found that substrates substituted with both electron-donating groups (alkyl, OMe, NH$_2$, or OPh) as well as electron-withdrawing groups (F, CF$_3$, CO$_2$Me, or *m*-NHMe) underwent the cyanation reaction smoothly. Low yield (**2o**, 37%) was obtained for the reaction of 3-chlorobenzonitrile, with benzonitrile generated as the main by-product from dehalogenative hydrogenolysis. Aryl chlorides containing two substitutents at different positions were also readily cyanated (**2p-2s**, 69–92%). It is noteworthy that various reducible functional groups such as imine, ester, amide, and olefin groups were tolerated in this system[48–50]. 1,2-Diaryl olefin and trisubstituted olefin were found compatible giving products with acceptable yields (**2x-2y**, 36–93%). In contrast, C=C bond of 1,1-diarylethene **1z** was reduced to give the product **2z** (55% yield). We next proceeded to assess the applicability of this method in modifying bioactive intermediates (**2aa-2ae**). And this method proved to be efficient in these cases with satisfactory yields obtained (57–94%). Specifically, cyanation of 2-chlorophenothiazine provided Cyamemazine precursor **2aa** in excellent yield (94%). Lipid lowering agent Clofibrate was successfully cyanated to produce **2ab** in modest yield (57%). Delightedly, cyanated products **2ac** and **2ad** were also obtained in 80% and 71% yields, respectively. In addition, transforming the anti-histaminic and anti-allergic Loratadine to nitrile **2ae** was achieved in decent yield (72%). Furthermore, compounds **2ad**, **2ae**, **2ad'**, and **2ae'** were evaluated for their antiproliferative activity against MCF-7 and Hela cells. As shown in Fig. 4,

compounds **2ad** and **2ae** showed moderate antiproliferative activities against these cell lines with IC$_{50}$ values ranging from 5.91 to 23.54 μM, while compounds **2ad'** and **2ae'** showed no activities toward tumor cell lines MCF-7 and Hela with IC$_{50}$ > 50 μM.

**Cyanation with other CO$_2$-derived CN sources.** Considering the easy access of many CO$_2$-derived compounds, we tested our reductive cyanation approach using other types of CN sources (Fig. 5). The results showed that the presence of NH$_2$ moiety is crucial for the cyanation reactivity. To our delight, excellent yield could be obtained under the optimized conditions with urea (97% yield). Here, it is interesting to stress that silyl isocyanate TMS-NCO is an appropriate CN source giving 74% yield of the desired product. The involvement of isocyanate intermediates has already been established in our previous work on the use of CO$_2$/NH$_3$ as the cyanating reagent[47].

**Substrate scope using urea as CN source.** Since urea is a product of dehydration of ammonia and carbon dioxide, we then considered to evaluate urea as the cyano source for substrate scope screening. As shown from Fig. 6, various functional groups were tolerated to producing the desired nitriles with excellent chemoselectivity. Generally, substrates with electron rich group in the *para*-position showed higher reactivity (**2e-2f**, **2af-2ag**). Lowering reaction temperature, the transformation proceeded similarly with substrates bearing electron deficient substituents affording acceptable yields (**2h-2i**, **2k**, 61–87%). Moreover, functional groups including hydroxyl, amino, methylamino, etc. at various positions on the aromatic ring remained unaltered even in the presence of excess reductant (**2l-2m**, **2ai-2ak**). The reactions proceeded smoothly for di-substituted substrates bearing methyl, alkoxy, amino, and trifluoromethyl groups (**2q-2s**, **2al-2am**).

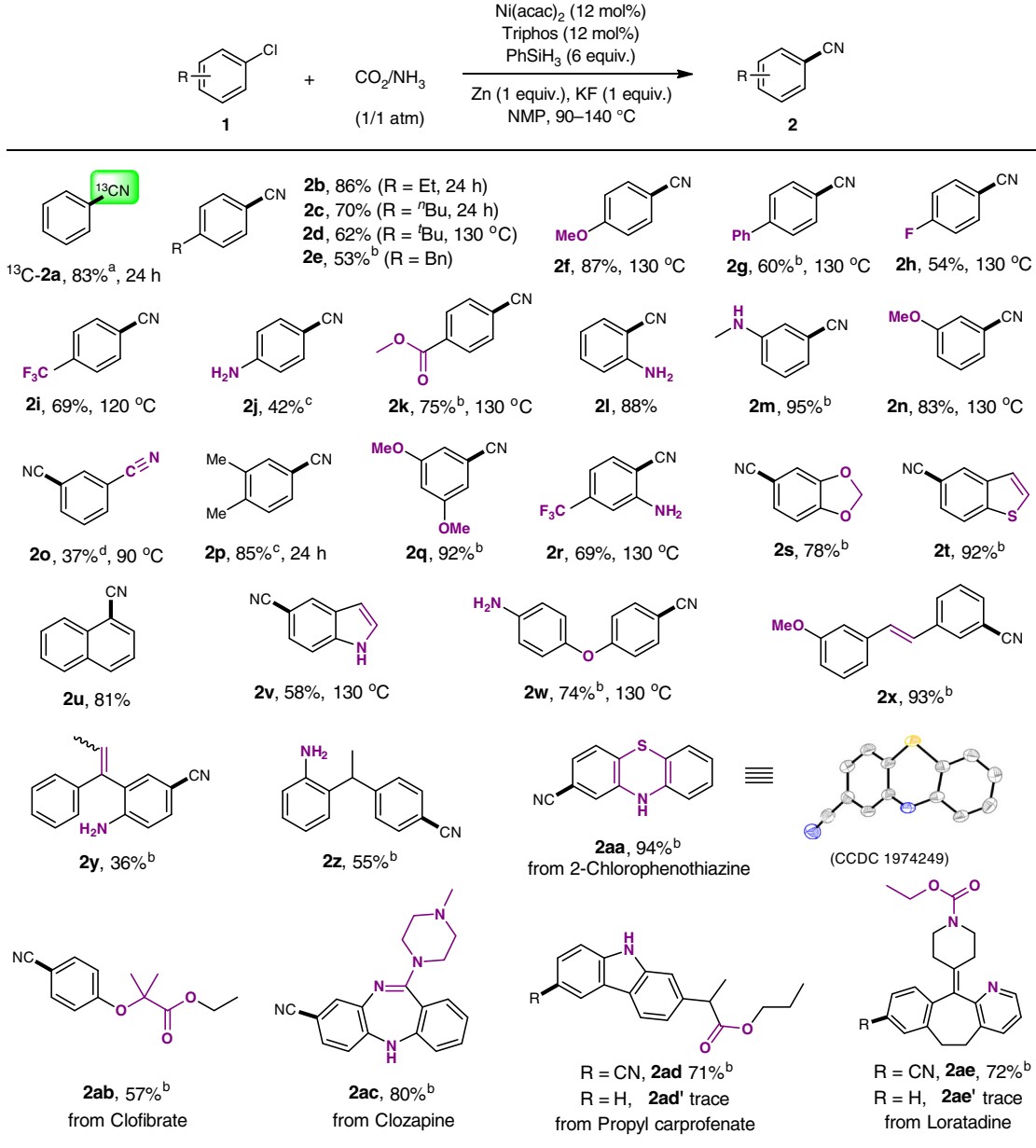

**Fig. 3 Applicability study using $CO_2/NH_3$.** Reaction conditions: **1** (0.125 mmol), $CO_2/NH_3$ (1/1 atm), NMP (0.5 mL), 20 h, GC yield. [a] $^{13}CO_2$ was used. [b] Isolated yields. [c] 15 mol% Ni(acac)$_2$, 15 mol% Triphos. [d] 10 mol% Ni(acac)$_2$, 10 mol% Triphos.

| Compd. | Anti proliferative activity, IC$_{50}$ (μM)[a] | |
|---|---|---|
| | MCF-7 | Hela |
| **2ad** | 19.35 ± 0.23 | 5.91 ± 0.13 |
| **2ae** | 23.54 ± 0.19 | 10.09 ± 0.21 |
| **2ad'** | >50 | >50 |
| **2ae'** | >50 | >50 |

**Fig. 4 In vitro antiproliferative activities of compounds 2ad, 2ae, 2ad', and 2ae'.** [a] Concentration that inhibits the proliferation of cancer cells by 50%. Cell proliferation was measured using the CCK8 assay after incubation with the compounds for 24 h. The mean values of three independent experiments ± SE are reported.

Various polycyclic aromatic skeletons, like naphthalene, indole, thianaphthene, and carbazole were also compatible, providing aromatic nitriles in moderate to excellent yield (**2t-2v**, **2an-2ax**, 54–97%). $^{13}$C-labeled nitrile compound was conveniently prepared in good yield using $^{13}$C-urea ($^{13}$C-**2ax**, 75%), and the product was characterized by NMR, HRMS, and single-crystal X-ray diffraction. Furthermore, heterocyclic substrates such as pyridine also showed gentle reactivity at 90 °C under optimized conditions. Direct reactions of chlorpromazine hydrochloride with urea delivered the corresponding product **2az** in modest yield with promazine as side product due to dehalogenation. To our delight, metoclopramide was transformed into target product in moderate yield despite the multiple substituents (**2ba**, 76%). Notably, the system was suitable for benzyl chlorides when dppp ligand was used instead of Triphos, leading to the corresponding benzyl nitriles **2bb-2bf** within a relatively short period of time (5 h; 35–85% yields). When cyclohexyl chloride or phenylethyl

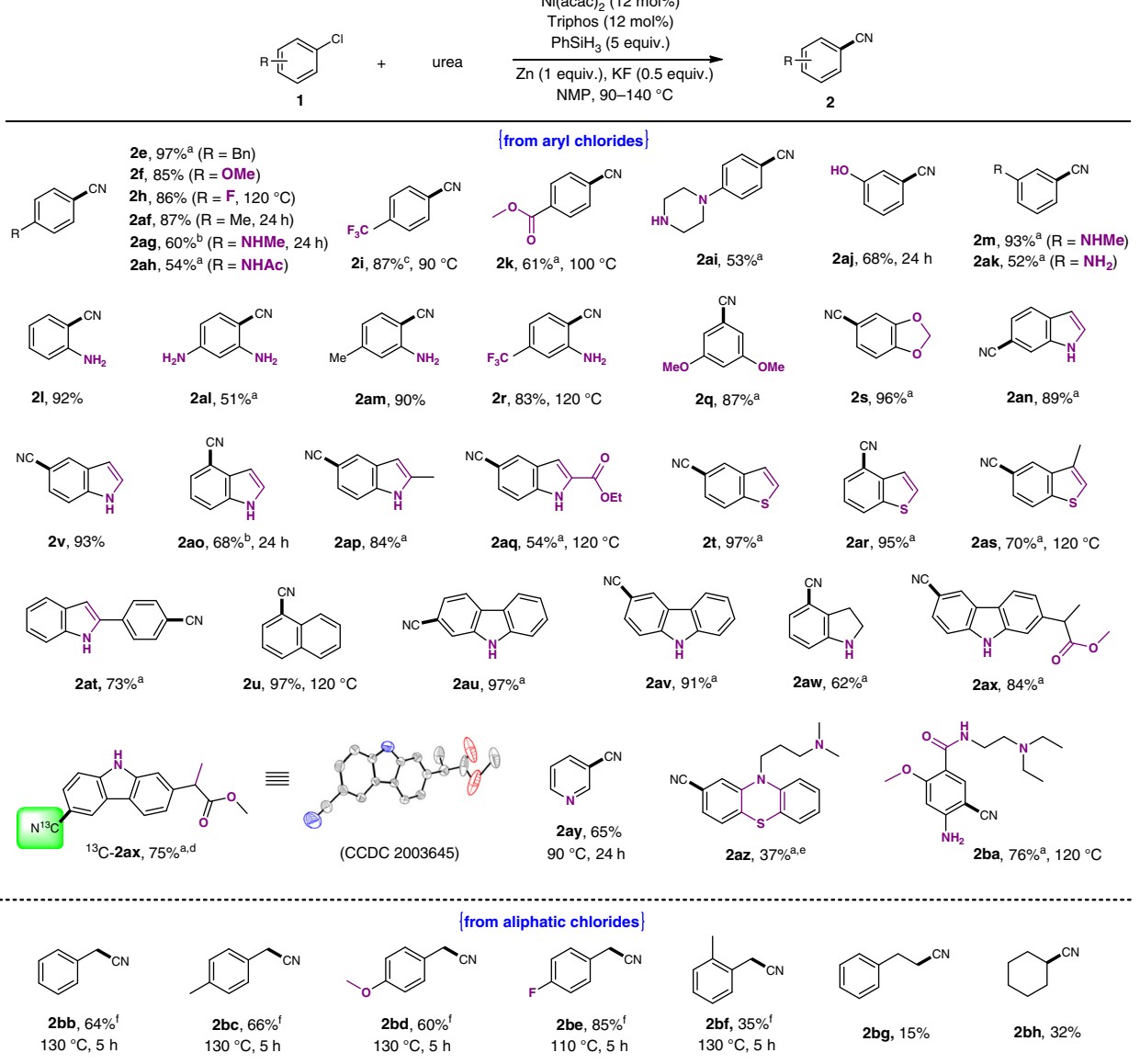

**Fig. 5 Catalytic cyanation with other related CN sources.** Reaction conditions: 0.3 mmol scale; GC yields. [a] 2 equiv. of PhSiH₃.

**Fig. 6 Substrate scope using urea.** Reaction conditions: **1** (0.3 mmol), urea (3 equiv.), NMP (1.0 mL), 20 h; GC yield. [a] Isolated yields. [b] 15 mol% Ni(acac)₂, 15 mol% Triphos. [c] K₃PO₄ was used instead of KF. [d] ¹³C-urea was used. [e] From chlorpromazine hydrochloride, 1.5 equiv. KF. [f] dppp was used instead of Triphos.

**Fig. 7 Stoichiometric reactions relevant to mechanism. a** Reactivity of Ar–Ni(I) generated in situ. **b** Reactivity of Ar–Ni(II) generated in situ. **c** Reactivity of Ar–Ni(II). **d** Reactivity of Ar–Ni(III) generated in situ.

chloride were used as the substrate, desired products were also obtained, albeit in low yields (15–32%).

**Mechanistic studies.** In order to examine the reactive species, we questioned whether the catalytic cyanation proceeded via an in situ generation of CN⁻. When tested with TMS-CN as cyanogen under standard conditions, cyanated product was undetected while substrate chlorobenzene was mostly recovered. This result suggests that CN⁻ is not involved in the reaction. In addition, when Et₃SiH was used as reductant, certain amounts of triethylsilyl isocyanate were detected even though cyanation product was not observed (Supplementary Fig. 4). Considering the thermolysis of urea to release isocyanic acid and ammonia, the formation of active silyl isocyanate intermediates is possible[51], which is consistent with the results using TMS-NCO as shown in Fig. 5. To further gather information on the active Ar–Ni species, several Ar–Ni complexes were prepared and used in stoichiometric reactions (Fig. 7). In these stoichiometric reactions, to avoid the possible unwanted F–Si interaction between KF and TMS-NCO, KF was added prior to TMS-NCO. First, product *o*-tolunitrile **2o-Me** was prepared from Ni(*o*-tolyl)(dppp), which was formed by transmetallation of (dppp)₂Ni(BF₄) with *o*-tolylmagnesium chloride and TMS-NCO (Fig. 7a). (Low yield of 6% for **2o-Me** is reasonable since *o*-chlorotoluene goes with dppp at standard conditions furnishing **2o-Me** in 11% yield.)

These results revealed that Ni(I) species play an important role in the catalytic cycle[52–56]. Exposure of PhNi(dppp)Cl prepared by oxidative addition of **1a** to Ni(dppp)₂, with TMS-NCO failed to

afford product **2a** whereas **2a** was obtained in 53% yield in the presence of PhSiH₃ (Fig. 7b). Such experiment suggests that Ni(I) species are responsible for the reactivity while Ni(II) species are not critical for the reaction. Direct reaction of *o*-tolylNi(dppf)Cl and TMS-NCO did not produce the desired product **2o-Me** (only 2% yield when adding PhSiH₃), thus supporting that Ni(I) aromatic species are possible key intermediates (Fig. 7c). Finally, the results of Ar–Ni(III) species generated in situ with TMS-NCO are in line with the pathway that Ni(I) species react with silyl isocyanates to form the C–C bond (Fig. 7d).

Based on these experimental results and previous reports[57–59], a possible reaction pathway is proposed as shown in Fig. 8. First, Ni(II)-precursors are reduced to Ni(0)-species **A** in the presence of silanes and Zn. This is followed by oxidative addition of the aryl chloride to **A** to form Ni(II) halide **B**, which is reduced by silane and Zn to afford highly nucleophilic Ni(I) intermediate **C**. Subsequently, silyl isocyanates formation[51] is followed by nickel–carbon insertion to generate presumably transient imidate species **D**. Further transformation yields the cyano product via a plausible 1,3-silyl N-to-O migration, whereby nickel siliconate intermediate **E** is released and further reduced by hydrosilane to regenerate species **A**. As discussed before, the formation of aryl-Ni(I) species is critical, while species **C** is stabilized by the tridentate phosphine ligand Triphos. The other advantage using Triphos is that it can act as the bidentate ligand favoring the formation of species **B** and **D** via addition reactions. The presence of Zn retarded the dehalogenative reduction process probably via inhibiting the formation of Ni(II)-hydride species[60]. The addition

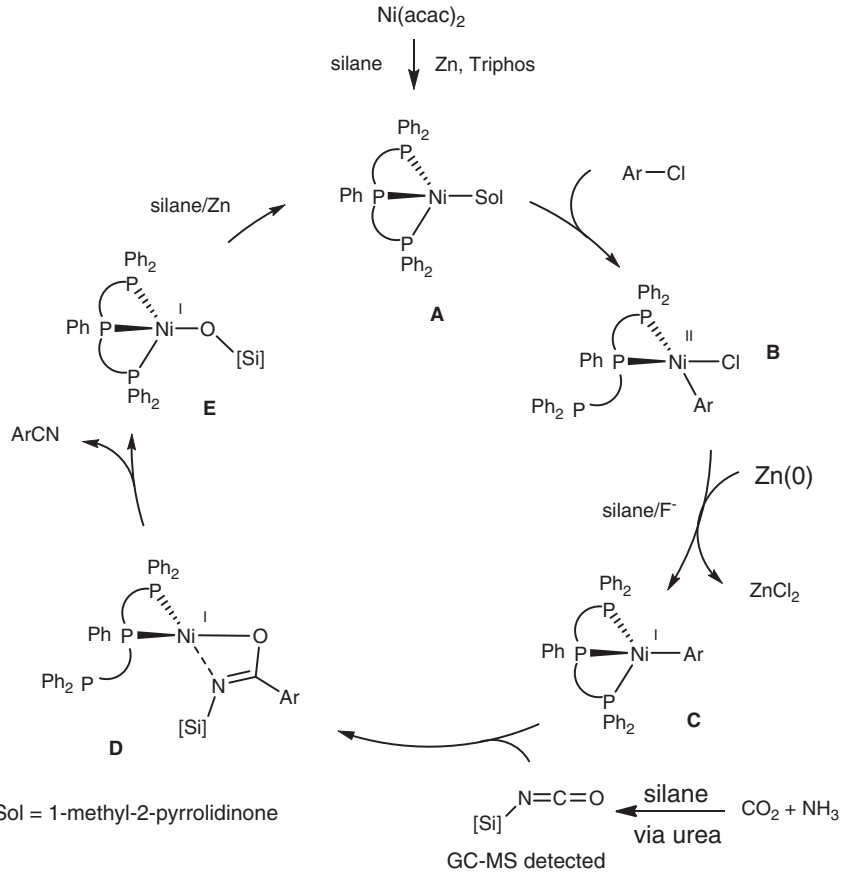

**Fig. 8 Plausible reaction mechanism.** Ni/phosphine-catalyzed reductive cyanation of organic chlorides using $CO_2/NH_3$.

of alkali metal chlorides such as LiCl can promote the reduction of Ni(II) complex to Ni(0) species by zinc flakes[61] and accordingly the addition of KF is likely to promote the reduction of Ni(II) species **B** to Ni(I) species **C**.

In summary, we have demonstrated the catalytic reductive cyanation of nonactivated aryl/heteroaryl and aliphatic chlorides with $CO_2/NH_3$ as cyano source. In the presence of nickel–triphos complexes, various organic chlorides were transformed into the desired aryl nitriles in moderate to excellent yields (up to 97%). Remarkably, diverse functional groups including nucleophilic hydroxyl, amino, unsaturated ester, amide, and olefin and heterocycle groups were tolerated. Cheap and bench-stable urea is also a proper CN source in this catalytic system, making it attractive to applications. Mechanistic studies demonstrate that Ar–Ni(I) species are responsible for the C–C coupling through the active silyl isocyanate intermediates.

## Methods

**General procedure for catalytic cyanation using $CO_2/NH_3$.** Under nitrogen atmosphere, Ni(acac)$_2$ (12 mol%, 0.015 mmol), Triphos (12 mol%, 0.015 mmol), KF (1.0 equiv., 0.125 mmol), Zn (1.0 equiv., 0.125 mmol), and a stirring bar were added into a 10 mL oven-dried sealed tube. Then NMP (0.5 mL), aryl chlorides (1.0 equiv., 0.125 mmol), and PhSiH$_3$ (6.0 equiv., 0.75 mmol) were injected by syringe. The tube was sealed and CO$_2$ (3.6 equiv., 10 mL) as well as NH$_3$ (3.6 equiv., 10 mL) were injected by syringe after N$_2$ was removed under vacuum. Then the mixture was stirred for 20 h in a preheated alloyed block. After the reaction finished, the tube was cooled to room temperature and the pressure was carefully released. The yield was measured by GC analysis or isolated by preparative thin-layer chromatography on silica gel plates eluting with PE/EtOAc.

## Data availability

The authors declare that the data supporting the findings of this study are available within the paper and its Supplementary Information files. Crystal structures have been deposited at the Cambridge Crystallographic Data Centre and allocated the deposition

numbers CCDC 1974249 (**2aa**), CCDC 2003645 ($^{13}$C-**2ax**) and 1974241 ((dppp)$_2$Ni (BF$_4$)). These data can be obtained free of charge from The Cambridge Crystallographic Data Centre via www.ccdc.cam.ac.uk/data_request/cif.Crystal data are also provided in Supplementary Information. All other data are available from the authors upon reasonable request.

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

## Acknowledgements
We thank the support from the National Natural Science Foundation of China (91745104, 21633013, and 21101109) and the National Science Foundation of Jiangsu Province (BK20180248).

## Author contributions
Y.L. and Y.D. conceived the project and wrote the manuscript. Y.D. performed the catalytic experiments and mechanistic studies. S.Z. performed the in vitro anti-proliferative activity experiments. P.Y. identify the structure of crystal. All authors contributed to the analysis and interpretation of the data.

## Competing interests
The authors declare no competing interests.
