## [Peer Review File · Nature Communications]

Reviewers' comments:

Reviewer #1 (Remarks to the Author):

Y. Dong et al. report a Ni-catalyzed reductive cyanation reaction of aromatic and aliphatic chlorides using urea, while the mechanistic studies showed good reasons that the reactions operate via the formation of urea from CO₂ and ammonia. A range of aryl chloride as well as benzyl chloride derivatives have been converted to the corresponding cyanide derivatives with moderate to good yields. Besides, the authors have also proposed a plausible mechanism based on control experiments and spectroscopic evidences. Some of Ar-CN products were tested for MCF-7 and HeLa cells, demonstrating biological activities of newly synthesized compounds as drug candidates.

The following comments should be addressed before making decision on the manuscript:

1. The title of the manuscript led this reviewer thinks that the most of reactions will be performed by CO₂ and ammonia, which is not the case in the reported cases (mainly urea mediated reactions). Therefore, the authors should redirect the focus of the manuscript by either changing the title or conducting additional experiments actually with CO₂ and ammonia.
2. Based on S. Chang's paper (J. AM. CHEM. SOC. 2010, 132, 10272) the authors must prove that the CN source is not derived from the -N-Me in the solvents since the reactions are all conducted in NMP or DMI with N-methyl functional group. This is quite important issue that must be resolved (See another paper utilizing urea and DMSO as sources for CN: Tetrahedron Letters Volume 54, 5250-5252, with Cu(II) and Ar-I)
3. In this sense, the authors should be able to observe the urea formation under optimized reaction conditions in the absence of substrates, to confirm the proposed reaction mechanism. (only TMS-isocyanate was observed by GCMS)
4. The substrate scope is impressive with urea as CN source. It is good to realize broad functional group tolerance with esters, thiazine, amidine, carbazole, carbamate, and pyridine (Figure 3). And the authors mentioned that the substrates containing unsaturated moieties are generally incompatible in reductive cyanation, without giving any references to the statement. Rather unsaturated compounds were found to be tolerated in Ni-catalyzed reductive cyanation as reported in ref 33 by Rousseaux. Also it is not clear if the authors intended to highlight functional group tolerance with various bioactive molecules as unsaturated hydrocarbons since there is only one compound actually having olefin moiety (2f) which can be reduced (but it's still not so easy since its tetrasubstituted).
5. Figure 4 should be replaced with higher resolution figure. Also, the authors should report control experiments with DMSO and parent compounds, original drug molecules, Ar-Cl and Ar-H to highlight the cyanation method.
6. Additional isotope experiments with ¹³CO₂ and/or ¹⁵NH₃ would be interesting to confirm the source of CN. There is no description regarding the formation of ¹³CN product (2a). What is the source of ¹³C? And why do authors observe the reduced yield?
7. Alkyl- and benzyl substrates showed in general lower yields. Is this reaction reversible? Is it possible that the reaction can participate reversible CN transfer reaction like Morandi's shuttle catalysis?

References to be cited:

A review on non-metallic cyanide sources for Ar-CN formation reactions. Angew. Chem. Int. Ed. 2012, 51, 11948 – 11959

Others comments:

Compound 2c has wrong HRMS calculated value therefore the observed value should be wrong as well. And the observed MS is exactly the same as the wrong calculated value. This make me wonder the integrity of the reported HRMS values.

In Table 1. The pressure of CO₂/NH₃ or the volume of the reaction container should be indicated

since the authors utilized 10/10 mL. Figure 3 indicates CO₂ (1atm) and NH₃ (1 atm) were used for the reactions, which require more information regarding the experimental details. The reaction of CO₂ and ammonia will immediately generate ammonium carbamate and carbamic acid, which can be isolated and tested as a reagent.

Reviewer #2 (Remarks to the Author):

On the basis of their previous work, the authors reported Ni-catalyzed cyanation of organic chlorides with CO₂ and NH₃ as the CN source. In addition, urea and silyl isocyanate were found to be used as CN source for the Ni-catalyzed cyanation of organic chlorides. The substrate scope and the reaction mechanism were well studied. The results obtained should be interesting to Catalytic/Organic Chemists. Thus, I recommend this manuscript for publication after minor revisions.

1. The cyanation process described in references 22 and 23 is a cyanides-free process rather than the in-situ generation of cyanide ion process. The authors should carefully check and classify the references
2. Usually, the fluoride was used for the activation of organosilane reagent. I noticed that the KF was added prior to TMS-NCO (Figure 7). The reason should be given. Whether KF plays other roles?
3. In Figure 7, the PhSiH₃ was used as reductant to reduce Ni(II) to Ni(I). However, according to the description of mechanism, the Ni(II) was reduced by Zn. It needs to revise.
4. If Zn could be used as a single reductant when TMS-NCO was used as the CN source?

Point-to-point Reply

Reviewers' comments:

Reviewer #1: Y. Dong et al. report a Ni-catalyzed reductive cyanation reaction of aromatic and aliphatic chlorides using urea, while the mechanistic studies showed good reasons that the reactions operate via the formation of urea from CO₂ and ammonia. A range of aryl chloride as well as benzyl chloride derivatives have been converted to the corresponding cyanide derivatives with moderate to good yields. Besides, the authors have also proposed a plausible mechanism based on control experiments and spectroscopic evidences. Some of Ar-CN products were tested for MCF-7 and Hela cells, demonstrating biological activities of newly synthesized compounds as drug candidates. The following comments should be addressed before making decision on the manuscript:

1. The title of the manuscript led this reviewer thinks that the most of reactions will be performed by CO₂ and ammonia, which is not the case in the reported cases (mainly urea mediated reactions). Therefore, the authors should redirect the focus of the manuscript by either changing the title or conducting additional experiments actually with CO₂ and ammonia.

Answer: Thank you very much for the comment. Following the suggestion, we have conducted additional substrate scope study experiments using CO₂/NH₃ and the results of **26 new examples** were added. As shown in the new Figure 3, the cyanation of 31 substrates were performed giving the desired products in moderate to excellent yields (36-95%). The new figure was incorporated into the manuscript including the *detailed discussion*: "An array of aryl chlorides bearing various substituents were well-tolerated to afford the desired nitrile products in moderate to high yields (Figure 3). ¹³C-labeled benzonitrile **13C-2a** was conveniently obtained using ¹³CO₂ (83% yield), which confirmed that the carbon source of cyano moiety was derived from CO₂ (see the Supporting Information). We found that substrates substituted with both electron-donating groups (alkyl, OMe, NH₂, or OPh) as well as electron-withdrawing groups (F, CF₃, CO₂Me, or *m*-NHMe) underwent the cyanation reaction smoothly. Low yield (**2o**, 37%) was obtained for the reaction of 3-chlorobenzonitrile, with benzonitrile generated as the main by-product from dehalogenative hydrogenolysis. Aryl chlorides containing two substituents at different positions were also readily cyanated (**2p-2s**, 69-92%). It is noteworthy that various reducible functional groups such as imine, ester, amide and olefin groups were tolerated in this system.⁴⁸⁻⁵⁰ 1,2-Diaryl olefin and trisubstituted olefin were found compatible giving products with acceptable yields (**2x-2y**, 36-93%). In contrast, C=C bond of 1,1-diarylethene **1z** was reduced to give the product **2z** (55% yield). We next proceeded to assess the applicability of this method in modifying bioactive intermediates (**2aa-2ae**). And this method proved to be efficient in these cases with satisfactory yields obtained (57%-94%). Specifically, cyanation of 2-chlorophenothiazine provided Cyamemazine precursor **2aa** in excellent yield (94%). Lipid lowering agent Clofibrate was successfully cyanated to produce **2ab** in modest yield (57%). Delightedly, cyanated products **2ac** and **2ad** were also obtained in 80% and 71% yields, respectively. In addition, transforming the anti-histaminic and anti-allergic Loratadine to nitrile **2ae** was achieved in

decent yield (72%).”

Figure 3. Applicability study using CO₂/NH₃.

2. Based on S. Chang's paper (J. AM. CHEM. SOC. 2010, 132, 10272) the authors must prove that the CN source is not derived from the -N-Me in the solvents since the reactions are all conducted in NMP or DMI with N-methyl functional group. This is quite important issue that must be resolved (See another paper utilizing urea and DMSO as sources for CN: Tetrahedron Letters Volume 54, 5250-5252, with Cu(II) and Ar-I)

Answer: Thank you for the suggestion. We have carried out the reaction using ¹³C-labelled CO₂ and the results of ¹³C NMR and GC-MS supported that the C source of CN is from CO₂ and not the N-methyl group of solvent molecules. Please see results of ¹³C-2a in Figures 3 and S2.

3. In this sense, the authors should be able to observe the urea formation under optimized reaction conditions in the absence of substrates, to confirm the proposed reaction mechanism. (only TMS-isocyanate was observed by GCMS)

Answer: Thank you for the suggestion. For the observation of urea formation, the reactions in the absence of halide substrates were carried out under optimized conditions using either $^{13}\text{C-CO}_2/\text{NH}_3$ or $^{13}\text{C-urea}$. The form of free urea $(\text{NH}_2)_2\text{CO}$ (159.8 ppm in $\text{DMSO-}d_6$) was not observed. However, in both cases, the peak at 157.8 ppm was observed immediately after the reaction. Considering the robustness of C-N and C=O bonds in urea molecule, this signal should be attributed to the formation of silylated urea, which was formed via dehydrogenative N-silylation of urea by PhSiH_3 under heating conditions. Consistently, the ^{13}C signal of N,N'-bis(trimethylsilyl)urea ($\text{DMSO-}d_6$) is at 157.95 ppm (AIST: Integrated Spectral Database System of Organic Compounds. (Data were obtained from the National Institute of Advanced Industrial Science and Technology (Japan))).

4. The substrate scope is impressive with urea as CN source. It is good to realize broad functional group tolerance with esters, thiazine, amidine, carbazole, carbamate, and pyridine (Figure 3). And the authors mentioned that the substrates containing unsaturated moieties are generally incompatible in reductive cyanation, without giving any references to the statement. Rather unsaturated compounds were found to be tolerated in Ni-catalyzed reductive cyanation as reported in ref 33 by Rousseaux.

Answer: Thank you for the suggestion. Selected reviews on on reduction of unsaturated compounds in the presence of hydrosilanes have been cited as ref. 48-50.

5. Also it is not clear if the authors intended to highlight functional group tolerance with various bioactive molecules as unsaturated hydrocarbons since there is only one compound actually having olefin moiety (2f) which can be reduced (but it's still not so easy since its tetrasubstituted).

Answer: Thank you for the suggestion.

(1) In the manuscript the sentence was changed to “We next proceeded to assess the applicability of this method in modifying bioactive intermediates (**2aa-2ae**)”.

(2) Besides, the results of three new substrates containing olefin groups were added in Figure 3. The discussion was added into the main text as “1,2-Diaryl olefin and trisubstituted olefin were found compatible giving products with acceptable yields (**2x-2y**, 36-93%). In contrast, C=C bond of 1,1-diarylethene **1z** was reduced to give the product **2z** (55% yield)”.

6. Figure 4 should be replaced with higher resolution figure. Also, the authors should report control experiments with DMSO and parent compounds, original drug molecules, Ar-Cl and Ar-H to highlight the cyanation method.

Answer: Thanks for the suggestion.

(1) Figure 4 has been replaced.

(2) Following the suggestion, we prepared the Ar-H compound derived from the drug molecules. And the results of these control experiments have been added into the manuscript, and the sentence “Furthermore, compounds **2ad**, **2ae**, **2ad'** and **2ae'** were evaluated for their

antiproliferative activity against MCF-7 and HeLa cells. As shown in Figure 4, compounds **2ad** and **2ae** showed moderate antiproliferative activities against these cell lines with IC_{50} values ranging from 5.91 to 23.54 μ M, while compounds **2ad'** and **2ae'** showed no activities toward tumor cell lines MCF-7 and HeLa with $IC_{50} > 50 \mu$ M." has been added.

7. Additional isotope experiments with $^{13}CO_2$ and/or $^{15}NH_3$ would be interesting to confirm the source of CN. There is no description regarding the formation of ^{13}CN product (**2al**). What is the source of ^{13}C ? And why do authors observe the reduced yield?

Answer: Thanks for the comment.

(1) Following the suggestion, we used $^{13}CO_2$ for the cyanation of **1a** and 83% yield of the desired isotope-labeled benzonitrile was obtained (as shown in the new Figure 3). The product was characterized by GC-MS and ^{13}C NMR.

(2) About the production of **2al** (new name: ^{13}C -**2ax**), we have 1) prepared the single crystal of this compound and had its structure characterized by X-ray diffraction; 2) added the description into the main text: " ^{13}C -labeled nitrile compound was conveniently prepared in good yield using ^{13}C -urea (^{13}C -**2ax**, 75%), and the product was characterized by NMR, HRMS and single-crystal X-ray diffraction"; 3) added the description into Figure 6: " ^{13}C -urea was used".

(3) About the reduced yield, there are two possible reasons: 1) it's known that $k^{12}C/k^{13}C$ KIE values could range from 1.01 to 1.04. In our system, cleavage of C-N and/or C-O bonds of silylated ureas and isocyanates might be relevant to slow steps during the catalytic cycle. Thus, slightly reduced yield for the ^{13}C -product is conceivable; 2) it might also be in part due to the different purity of the purchased urea and ^{13}C -urea.

8. Alkyl- and benzyl substrates showed in general lower yields. Is this reaction reversible? Is it possible that the reaction can participate reversible CN transfer reaction like Morandi's shuttle catalysis?

Answer: Thanks for the comment. The result about low yields is due to the significant dehalogenative hydrogenolysis of aliphatic chlorides to generate the alkane by-products. Besides, isocyanates were tested and proposed as the reaction intermediates. When using TMS-CN as the CN source, under otherwise the same conditions, no desired cyanated product was detected (Figure 5). Thus, these results suggest that this system is not relevant to the Morandi's shuttle catalysis.

9. References to be cited: 1. A review on non-metallic cyanide sources for Ar-CN formation reactions. *Angew. Chem. Int. Ed.* 2012, 51, 11948-11959

Answer: This ref. has been cited as ref. 7.

10. Compound **2c** has wrong HRMS calculated value therefore the observed value should be wrong as well. And the observed MS is exactly the same as the wrong calculated value. This make me wonder the integrity of the reported HRMS values.

Answer: Thanks for the comment. Firstly, the yield of **2c** (new name: **2aa**) is correct and this compound was also characterized by single-crystal X-ray diffraction. About HRMS data, we

checked the HRMS spectrum of **2c** (new name: **2aa**) and found that a typo was made when inputting the value into the SI: HRMS (ESI, m/z): “calcd for $C_{13}H_8N_2NaS^+$ $[M+Na]^+$: 207.0300, found: 207.0300”. It should be: “calcd for $C_{13}H_8N_2NaS^+$ $[M+Na]^+$: 247.0300, found: 247.0300”. Now it is corrected and the spectrum of **2c** (new name: **2aa**) is shown below. And, we have checked other HRMS values carefully, too.

11. In Table 1. The pressure of CO_2/NH_3 or the volume of the reaction container should be indicated since the authors utilized 10/10 mL. Figure 3 indicates CO_2 (1atm) and NH_3 (1 atm) were used for the reactions, which require more information regarding the experimental details. The reaction of CO_2 and ammonia will immediately generate ammonium carbamate and carbamic acid, which can be isolated and tested as a reagent.

Answer: Thanks for the suggestion.

(1) Following the comment, we decided to always use pressure for CO_2/NH_3 (1 atm and 1 atm, respectively). Accordingly, detailed description was added into the part of **experimental procedures** in the SI.

(2) The immediate generation of ammonium carbamate was observed and could be collected as white solid on the upper inside wall of reaction vessels, which is highly dissolvable in H_2O . We have tested the use of ammonium carbamate and methyl carbamate as the CN source and moderate to good yields were obtained as 43% and 77%, respectively (Figure 5).

Reviewer #2:

On the basis of their previous work, the authors reported Ni-catalyzed cyanation of organic chlorides with CO₂ and NH₃ as the CN source. In addition, urea and silyl isocyanate were found to be used as CN source for the Ni-catalyzed cyanation of organic chlorides. The substrate scope and the reaction mechanism were well studied. The results obtained should be interesting to Catalytic/Organic Chemists. Thus, I recommend this manuscript for publication after minor revisions.

1. The cyanation process described in references 22 and 23 is a cyanides-free process rather than the in-situ generation of cyanide ion process. The authors should carefully check and classify the references

Answer: Thanks a lot for the suggestion. We have carefully checked all the references again. References 22 and 23 have been moved to the end of this sentence “elegant development of cyanide-free methods involving diverse mechanisms to circumvent this challenge has been achieved including β-H elimination-facilitated cyanation” in paragraph 3.

2. Usually, the fluoride was used for the activation of organosilane reagent. I noticed that the KF was added prior to TMS-NCO (Figure 7). The reason should be given. Whether KF plays other roles?

Answer: Thanks a lot for the suggestion.

(1) The main reason to add KF prior to TMS-NCO is to avoid the possible F-Si interaction, which might retard the desired promotion effect in control experiments. This sentence is added into the main text: “In these stoichiometric reactions, to avoid the possible unwanted F-Si interaction between KF and TMS-NCO, KF was added prior to TMS-NCO.”.

(2) KF might play two roles: 1) as the base; 2) as the counter ion to stabilize the key Ni-species (For fluoride anion effect on metal center to suppress the Ni-catalyzed homocoupling, please see the ref. 12 of this paper: Takuji Hatakeyama and Masaharu Nakamura, J. Am. Chem. Soc. 2007, 129, 9844-9845). For the reaction of Figure 7b, we tested different conditions to further understand the role of KF. Without KF, the yield of 12% was obtained. When adding K₃PO₄ instead of KF, the yield of the product increased to 36%. And, the addition sequence indeed influences the reactivity. Slightly lower yield of 44% (compared to the yield of 53% in Figure 7b) was obtained when KF and TMS-NCO were added at the same time.

3. In Figure 7, the PhSiH₃ was used as reductant to reduce Ni(II) to Ni(I). However, according to the description of mechanism, the Ni(II) was reduced by Zn. It needs to revise.

Answer: Thanks a lot for the comment. We have revised Figure 8 and the description of mechanism accordingly. In Figure 8, from species **B** to **C**, the presence of silane was added. We changed the sentence to “This is followed by oxidative addition of the aryl chloride to **A**

to form Ni(II) halide **B**, which is reduced by silane and Zn to afford highly nucleophilic Ni(I) intermediate **C**.” in the main text.

4. If Zn could be used as a single reductant when TMS-NCO was used as the CN source?

Answer: Thanks a lot for the comment. We have tried Ni-catalyzed cyanation of 4-chlorotoluene with TMS-NCO using Zn as the reductant in the absence of hydrosilanes. No desired nitrile product was detected after the reaction. And, the generation of significant amounts of toluene (from dehalogenation) and dimethylbiphenyl (from reductive coupling) were observed.

REVIEWERS' COMMENTS:

Reviewer #1 (Remarks to the Author):

The authors demonstrate the main point of the concept very nicely.
I am happy to see all the new experimental results supporting the idea.